# Interaction-aware Dynamic 3D Gaze Estimation in Videos

**Chenyi Kuang**[1]                                                            KUANGC2@RPI.EDU
**Jeffrey O. Kephart**[2]                                                    KEPHART@US.IBM.COM
**Qiang Ji**[1]                                                                    JIQ@RPI.EDU
[1]*Rensselaer Polytechnic Institute, 110 8th St, Troy, NY 12180, USA*
[2]*IBM Thomas J. Watson Research Ctr., 1101 Kitchawan Rd, Yorktown Heights, NY 10598, USA*

## Abstract

Human gaze in in-the-wild and outdoor human activities is a continuous and dynamic process that is driven by the anatomical eye movements such as fixations, saccades and smooth pursuit. However, learning gaze dynamics in videos remains as a challenging task as annotating human gaze in videos is labor-expensive. In this paper, we propose a novel method for dynamic 3D gaze estimation in videos by utilizing the human interaction labels. Our model contains a temporal gaze estimator which is built upon Autoregressive Transformer structures. Besides, our model learns the spatial relationship of gaze among multiple subjects, by constructing a Human Interaction Graph from predicted gaze and update the gaze feature with a structure-aware Transformer. Our model predict future gaze conditioned on historical gaze and the gaze interactions in an autoregressive manner. We propose a multi-state training algorithm to alternately update the Interaction module and dynamic gaze estimation module, when training on a mixture of labeled and unlabeled sequences. We show significant improvements in both within-domain gaze estimation accuracy and cross-domain generalization on the physically-unconstrained gaze estimation benchmark.

**Keywords:** 3D Gaze estimation, Human gaze interaction, Gaze dynamics.

## 1. Introduction

Eye gaze is an important cue for human behaviour and attention analysis. With the growing popularity in interactive applications such as AR/VR, 3D avatar animation, human-computer interaction and driver behaviour monitoring, automatic gaze estimation methods are proposed to regress 3D gaze directions from eye images. More recently, with the enrichment of large scale gaze datasets Kellnhofer et al. (2019); Zhang et al. (2020); Funes Mora et al. (2014); Fischer et al. (2018), deep learning models have been fully utilized to regress gaze from images captured in different environments. Despite the progresses in image-based gaze estimation, gaze dynamics learning has not yet been fully explored. First, it is difficult to capture eye movement dynamics accurately in videos when the subject has frequent body or head movement, which may cause blur or occlusion in the eye region. Second, the dynamic eyeball movement in a video dataset may be elicited by specific tasks or scenarios, so it's questionable if such models can well generalize to other dataset. Finally, annotating gaze frame-by-frame for videos can be time consuming and labour-intensive and deep learning models may suffer from inadequate training labels.

Several methods have been proposed in previous researches to model eyeball movement or gaze dynamics in videos. A recurrent CNN is proposed by Palmero et al. (2018) to model

the temporal dependency of 3D gaze in a sequence, which predict the gaze direction only in the last frame. Other recurrent modules have been considered, such as GRU (Park et al. (2020)) and LSTM (Kellnhofer et al. (2019),Palmero Cantarino et al. (2020)). Besides, Nonaka et al. (2022) proposes a dynamic framework, by formulating probabilistic gaze estimation given temporal estimation of head and body orientation. The idea of using head & body orientation likelihoods to model the temporal prior of gaze in (Nonaka et al. (2022)) reveals the advantage of dynamic gaze estimation, which is to reasonably infer the gaze even when the eye region is invisible due to occlusion or low image resolution.

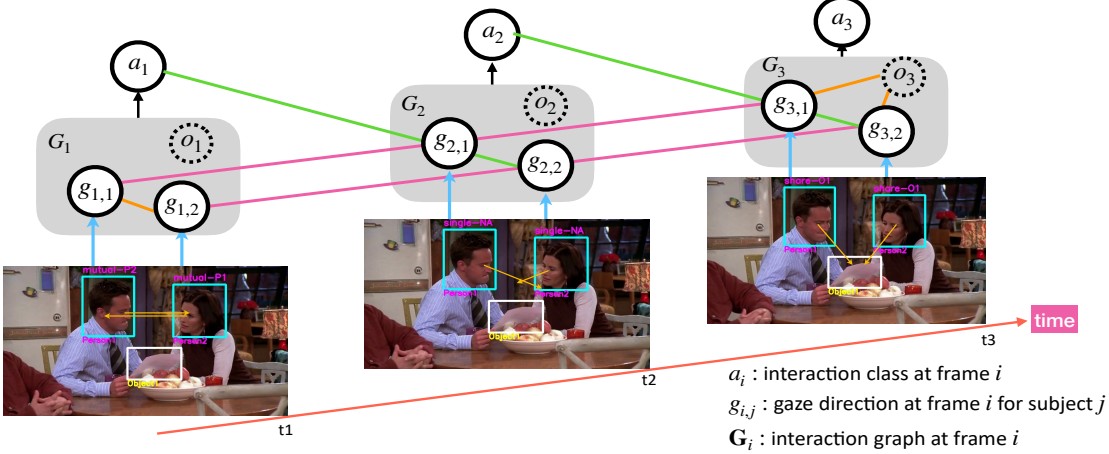

Figure 1: Spatial and Temporal relationships among gaze directions and human interaction classes.

Due to the difficulties in creating fully-annotated video datasets for gaze estimation, researchers seek for "secondary" labels which are easier to acquire to refine the gaze estimator. For example, Park et al. (2020); Wang et al. (2019) both incorporate point of gaze (PoG) estimation in a gaze estimation framework, where the subject conducts natural eye movements following a moving target in a video. The eye gaze estimation is refined jointly with the PoG estimation process through learning temporal relationships, without requiring ground-truth gaze annotations. In addition to PoG labels, Kothari et al. (2021) proposes a novel weakly-supervised framework for learning 3D gaze from videos, where people are "Looking At Each Other" (LAEO). The LAEO labels are formulated as geometric gaze constraints to supervise the training process. Kothari et al. (2021) successfully proved that utilizing the human interaction information in social scenarios can significantly improve the within-domain accuracy and cross-domain generalization ability. However, Kothari et al. (2021) only utilize the specific human interaction type (i.e., looking at each other or not), and fail to consider the variety of human interaction activities. The temporal relationships learned from LAEO labels can be too weak to represent the gaze dynamics in a social scenario.

In this work, we propose a novel model to enhance the dynamic 3D gaze estimation in videos by learning the state and dynamic transitions of various human gaze interaction activities. We refer to the atomic-level and event level gaze communication activities defined

by Fan et al. (2019) and model the dynamic transitions among six types of gaze communication activities, including {*single, mutual, avert, refer, follow, share*}. We use the predicted gaze direction and subject location information to construct a Human Interaction Graph and utilize the historical state to infer the current state. We first train our gaze estimator on Gaze360 (Kellnhofer et al. (2019)), then jointly optimize the dynamic gaze estimator and the interaction learning module on VACATION dataset (Fan et al. (2019)), without using additional gaze annotations, as shown in Fig. 1. Our contribution includes:

- We propose a dynamic 3D gaze estimation framework for learning 3D gaze from human interaction videos. Compared to previous work which only explore LAEO cases, we consider a variety of human interactions and the transitions between two interaction states. To our knowledge, this is the first attempt to use multiple human interaction activities to enhance dynamic 3D gaze learning.

- To effectively model the interaction between subjects with gaze, we propose a spatio-temporal model through combining Human-Interaction Graph with a Transformer-based spatial and temporal module, to jointly capture the spatial and temporal relationships of human gaze movement.

- We use predicted gaze to construct the Human Interaction Graph and interaction classification loss and develop a multi-stage training algorithm to alternately update the interaction module and gaze module. The results of within- and cross-domain evaluation shows that the human interaction learning can effectively enhance the gaze estimator.

## 2. Related Works

### 2.1. Dynamic 3D Gaze Estimation

Fully supervised learning based gaze estimation methods have achieved impressive within-domain performances on static images, such as Cheng et al. (2020); Chen and Shi (2018); Fischer et al. (2018); Zhang et al. (2017). However, dynamic gaze estimation has not been extensively explored due to lack of fully-annotated gaze videos. With the sequence-based gaze labels from recently-published dataset EyeDiap Funes Mora et al. (2014) or Gaze360 Kellnhofer et al. (2019), a few temporal gaze estimation models have been proposed to predict eye gaze direction from a image sequence. Palmero et al. (2018) proposed a multimodal recurrent CNN framework that feed the concatenated static feature of each frame into a recurrent module for predicting the 3D gaze direction of the last frame in the sequence. Similarly, Kellnhofer et al. (2019) have proposed to use a bidirectional LSTM to encode the contextual information in temporal domain and predict gaze for the central frame. Such fully-supervised models may fail to generalize to different datasets under various environments. In the meantime, multiple researches have been conducted to explore other sources of labels that can help refine or provide weak supervision to the gaze model. Wang et al. (2019) collected a dataset which records human eye images and the ground-truth gaze positions on a screen while subjects are browsing websites or watching videos. A dynamic gaze transition network is proposed to capture the transitions of different eye movements in temporal domain, then refine the static gaze predictions with learned dynamics. Park et al.

(2020) constructed a large-scale video-based eye tracking dataset with ground-truth Point of Gaze (PoG) on a screen, followed by a recurrent module that performs PoG refinement task on video data. Utilizing auxiliary information from body or head pose have been used for unconstrained gaze estimation. Nonaka et al. (2022) formulated a Bayesian gaze estimation framework given temporal estimates of 3D head and body orientations, which can be reliably estimated from a far distance. The above mentioned methods encode eye gaze dynamics based on the motion prior for a single subject. As far as we know, learning eye movement dynamics from multi-subject interaction videos are not fully explored.

## 2.2. Gaze Target Estimation and Human Gaze Interaction

Eye gaze is an essential non-verbal clue for human activity, intention and communication analysis. Compared to the time-consuming 3D gaze direction annotation process, labeling gaze targets in a image/video is more straightforward to undertake. Recasens et al. (2015) first defined the gaze following task, which is to predict the location that each person in a scene is looking at from a single image. Chong et al. (2018) addressed a more challenging problem of estimating general human visual attention, which handles special cases such as out-of-frame gaze targets and looking-at-camera gaze. These datasets and techniques have been extended to video domain using temporal networks, such as the video-based gaze following framework proposed by Recasens et al. (2017) and video-based visual targets analysis proposed by Chong et al. (2020). Fang et al. (2021) enhance the gaze target estimation model by exploiting 3D scene context, including the 3D gaze direction, 3D head pose and scene depth. Estimating gaze target is useful in analyzing human visual attention but does not provide direct information about 3D gaze direction.

Human gaze interaction provide weak supervision to gaze direction learning when there exists multiple people in the scene. One type of useful weak supervision is mutual gaze, where two people are looking at each other (LAEO). Marin-Jimenez et al. (2014, 2019) formulated detecting LAEO between human as a binary classification task. Fan et al. (2019) extended the scope to multi-agent gaze communication behaviours in realistic social scene and distinguished six types of atomic-level gaze interactions. Compared to LAEO cases, Fan et al. (2019) further considered long-term gaze interaction dynamics and divided temporal compositions of atomic gazes into five classes of events, including {Non-Communicative, Mutual Gaze, Gaze Aversion, Gaze Following, Joint Attention}.

Kothari et al. (2021) creatively utilized LAEO labels from a web-video dataset for weakly supervised 3D gaze learning. The LAEO labels between a pair of subjects provide a strong geometric constraint that their gaze should be in opposite direction. Based on the LAEO constraint, Kothari et al. (2021) formulated pseudo gaze labels on LAEO pairs, which can guide the gaze model learning when one subject is face away from camera and the face/eye region is not visible. However, as we know, human interactions are usually dynamic activities, focusing on LAEO cases will largely ignore eye movement dynamics. To our knowledge, multi-class human interaction labels scene have not been used in supervising gaze estimation models. Besides, we are also the first work to learn temporal dependency of 3D gaze direction from interaction transitions.

## 3. Method

### 3.1. Problem Formulation

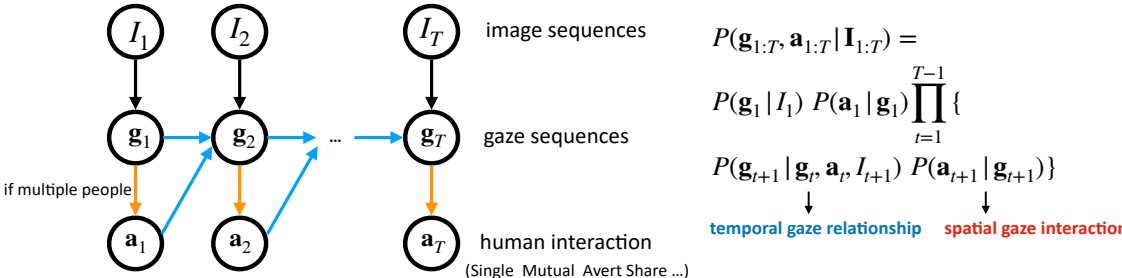

Figure 2: Problem formulation for Interaction-Aware dynamic gaze estimation.

We target at dynamic 3D gaze estimation in videos by modeling both temporal and spatial gaze relationships, as shown in Fig. 2. For a given image sequence $\boldsymbol{I} = \{I_1, \cdots, I_T\}$ that contains gaze communications among two or more people, we want to predict the gaze $\boldsymbol{g}$ and their interaction category $\boldsymbol{a}$ for each frame, by modeling the spatio-temporal gaze relationships. Assuming that the human gaze interactions can be fully inferred from gaze and can affect future gaze direction, the problem is formulated as:

$$P(\mathbf{g}_{1:T}, \mathbf{a}_{1:T}|\mathbf{I}_{1:T}) = P(\mathbf{g}_1|I_1) \, P(\mathbf{a}_1|\mathbf{g}_1) \prod_{t=1}^{T-1} \{P(\mathbf{g}_{t+1}|\mathbf{g}_t, \mathbf{a}_t, I_{t+1}) \, P(\mathbf{a}_{t+1}|\mathbf{g}_{t+1})\} \tag{1}$$

The gaze $\boldsymbol{g}_t \in \mathbb{R}^{N_t \times 3}$ contains multiple unit gaze vectors for $N_t$ subjects in frame $t$, i.e., $\boldsymbol{g}_t = [g_{t,1}, \cdots, g_{t,N_t}]$. The gaze interaction variable $\boldsymbol{a}_t \in \mathbb{Z}^{N_t \times 6}$ represents the one-hot interaction category vector for six types of interactions, including {*single, mutual, avert, refer, follow, share*} and $\boldsymbol{a}_t = [a_{t,1}, \cdots, a_{t,N_t}]$. In Eq. 1, we model $P(\mathbf{a}_t|\mathbf{g}_t)$ with learning a Human Interaction Graph $G_t$ from $\mathbf{g}_t$; then we model $P(\mathbf{g}_{t+1}|\mathbf{g}_t, \mathbf{a}_t, I_{t+1})$ with a Structure-aware Transformer and an Autoregressive Temporal Transformer.

### 3.2. Method Overview

We show the overview of the proposed framework in Fig. 3. Given a sequence of subject head images $\boldsymbol{I} = \{I_1, \cdots, I_T\}$, our network is defined as $\mathcal{F}(\cdot)$ and contains five sets of network parameters, including the ResNet-18 feature extractor $\mathcal{F}_{\Theta_1}$, the Structure-aware Transformer layer $\mathcal{F}_{\Theta_2}$ for updating the gaze feature with multi-subject interaction information, a temporal model $\mathcal{F}_{\Theta_3}$ built with Autoregressive Transformer Layers, a Fully-Connected (FC) layer $\mathcal{F}_{\Theta_4}$ for regressing the gaze vector and an interaction classifier $\mathcal{F}_{\Theta_5}$ for distinguishing the interaction category from gaze.

The prediction is run in an autoregressive manner, as formulated in Eq. 1. The predicted gaze $\mathbf{g}_t$ at time $t$ will be used to construct the edges of a Human-Interaction graph $G_t$, then we use s Structure-aware Transformer and a Temporal Transformer to model the distribution of future gaze, given the historical gaze and the gaze interaction. In the Human-Interaction Module, we first construct a human-interaction graph $\mathcal{G}_t = (V_t, E_t)$ using gaze and position

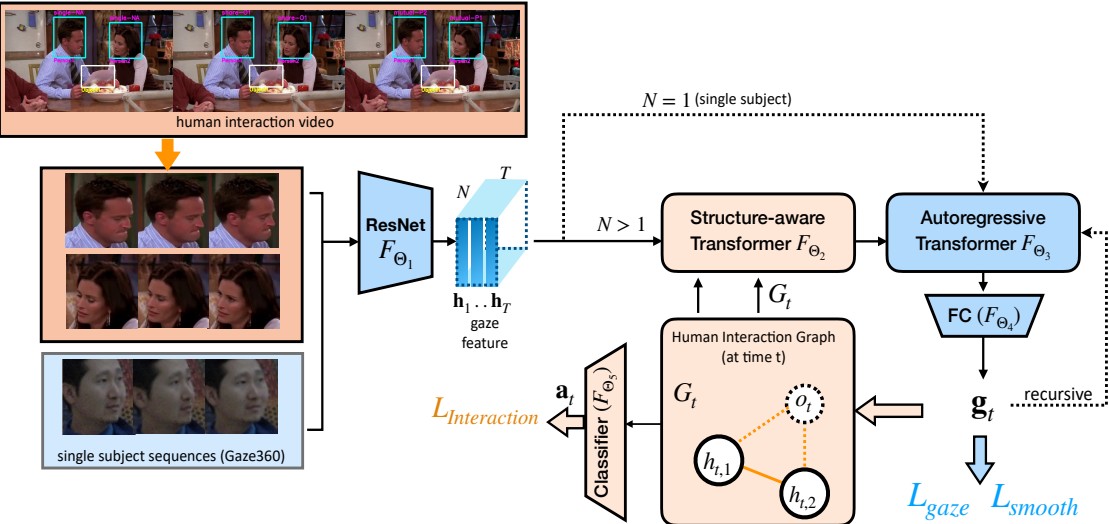

Figure 3: Overview of our method.

information for each frame $t$. Then we update the gaze representations in both spatial and temporal scale. The updated gaze feature are feed into prediction head and predict human interaction categories for each subject.

In Section 3.3, we show details about building the Human-Interaction Graph from predicted gaze. In Section 3.4, we describe the structure-aware transformer for learning the gaze interaction relationships. In Section 3.5, we describe the temporal module for predicting future gaze. In Section 3.6, we define a interaction classifier using the updated gaze features. We propose a multi-state training strategy to refine the dynamic estimate module. We define the loss function for training in Section 3.7 and describe the training algorithm in Section 3.8.

### 3.3. Human Interaction Graph

We use human interaction videos and labels from VACATION Fan et al. (2019) dataset to perform joint learning of 3D gaze dynamics and gaze interaction dynamics. The bounding boxes of all subjects and objects involved in the communication scene are provided. With the predicted gaze and bounding box locations, we propose to construct a human-interaction graph $\boldsymbol{G}_t = (\boldsymbol{V}_t, \boldsymbol{E}_t)$ for every frame $t$, where the nodes $\boldsymbol{V}_t$ can be further split into subjects nodes $\boldsymbol{V}_t^s$ and objects nodes $\boldsymbol{V}_t^o$. There can be a directed edge $e^{(ij)} \in \mathbf{E}_t$ from node $v^i$ to node $v^j$, indicating the subject $i$ is looking at another subject or an object.

At frame $t$, given the predicted gaze direction $\boldsymbol{g}_t = \{g_{t,i}\}_{i=1}^N$ and gaze uncertainty $\boldsymbol{\sigma}_t = \{\sigma_{t,i}\}_{i=1}^N$, we calculate the inter-activeness score for each subject-subject and subject-object pair, defined as below. For subject $i$, we first generate a 2D gaze attention map $M_i$ by calculating the angular difference $\theta$ between the gaze vector and the vector from one image pixel to the head center position $[d_{i,x}, d_{i,y}]$, formulated as

$$\theta_i(x,y) = \arccos\left(\frac{x - d_{i,x}, y - d_{i,y} \cdot (g_x, g_y)}{\|(x - d_{i,x}, y - d_{i,y})\|_2 \cdot \|g_x, g_y\|_2}\right) \text{ and } M_i(x,y) = \max\left(1 - \frac{\theta_{x,y}}{\alpha \sigma_i}, 0\right) \quad (2)$$

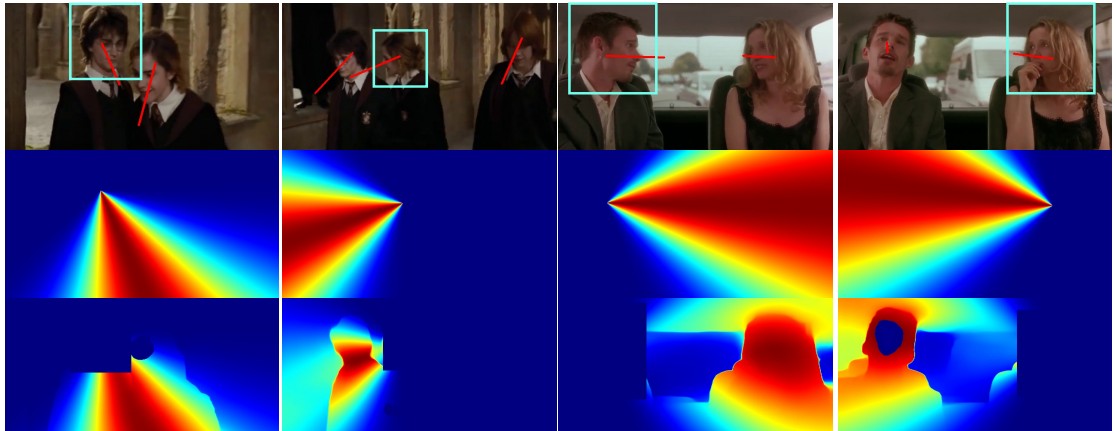

Figure 4: Visualization of the gaze view field based on the predicted gaze. First row: the predicted gaze direction. Second row: the 2D gaze field map generated based on the 2D gaze direction and the depth information in Eq. 2. Third row: the optimized gaze field map after applying depth rebasing by adding depth threshold on the gaze target region.

where $\alpha > 1$ is a hyperparameter, deciding the scope of gaze view field. Then we follow the depth rebasing method in Fang et al. (2021) to optimize the gaze field using the relative depth information. Examples of the initial gaze view field and optimized ones are shown in Fig. 4. At last, we calculate the interaction score $c_i$ from subject $i$ to other subjects or objects as:

$$c_{ij} = mean([M_i(x,y)], \forall (x,y) \text{ in bbox}_j) \tag{3}$$

The interaction score $c_{ij} \leq 1$ represents the probability of the edge connectivity. When the ground truth edge connectivity $e_{ij}^{gt} \in \{0,1\}$ is given, we can compute a graph structure loss to refine the gaze direction, which is defined below.

$$L_{G,t} = \frac{1}{|V_t^s|(|V_t^s| + |V_t^0|)} \sum_i^{|V_t^s|} \sum_j^{|V_t^s + V_t^0|} -e_{ij}^{gt} \log c_{ij} - (1 - e_{ij}^{gt}) log(1 - c_{ij})$$

$$L_G = \frac{1}{T} \sum_t^T L_{G,t} \tag{4}$$

### 3.4. Structure-aware Transformer

To model the distribution of $P(\mathbf{g}_{t+1}|\mathbf{g}_t, \mathbf{a}_t, I_{t+1})$, we propose to build a sequential model consists of a structure-aware Transformer and an autoregressive transformer that can generate gaze prediction conditioned on previous gaze, human interaction and image feature. The "structure-aware" transformer integrates gaze interaction information for generating

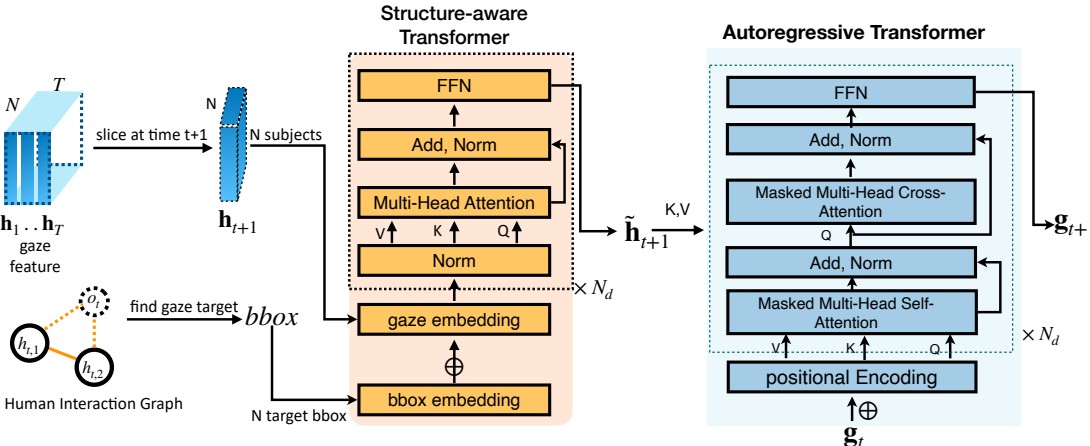

Figure 5: Detailed network structure of the Spatial and Temporal Transformer.

interaction-aware gaze embedding. As shown in Fig. 5, given the extracted image feature $\boldsymbol{h}_t$ in a frame for $N$ subjects, we project the feature vector to gaze feature embedding space to generate image tokens. From the human interaction graph $G_t$, we find the gaze target with the highest interaction score and feed the bounding box coordinates into the "bbox embedding" layer to generate target position tokens. The image token and target position tokens are concatenated as the input of transformer layers. The processing steps of the transformer model are described below.

$$
\begin{aligned}
&\text{Input: image feature } \boldsymbol{h}_{t+1}, \text{Interaction Graph: } G_{t-1} \\
&\text{tokens: } \boldsymbol{q}_{t+1,k} = [\text{MLP}(h_{t+1,k}); \text{MLP}(\text{bbox}_{t,k})], k = 1, \cdots, N \\
&\text{self-attn: } \boldsymbol{q}_{t+1}^{(l)} = \text{LN}(\text{MHS}(Q^{\boldsymbol{q}_{t+1}^{(l-1)}}, K^{\boldsymbol{q}_{l-1}}, V^{\boldsymbol{q}_{t+1}^{(l-1)}}) + \boldsymbol{q}_{t+1}^{(l-1)}) \\
&\text{FFN: } \boldsymbol{q}_{t+1}^{(l)} = \text{FFN}(\boldsymbol{q}_{t+1}^{(l)}), l = 1, \cdots, N_d \\
&\text{Output: } \tilde{\boldsymbol{h}}_{t+1} = \boldsymbol{q}_t^{(N_d)}
\end{aligned}
\tag{5}
$$

The output $\tilde{\boldsymbol{h}}_{t+1}$ is the updated gaze feature embedding at frame $t+1$, which will be used as input of the Temporal model. We introduce the details in next section.

### 3.5. Autoregressive Transformer for Temporal Relationship Learning

In stead of only predicting gaze direction for the central frame in a sequence like (Kellnhofer et al. (2019); Kothari et al. (2021)), our gaze module predicts gaze for each frame in the sequence as we will need gaze direction to learn human interaction dynamics. A shown in Fig. 3, the concatenated feature sequences are feed into the Temporal Module, which is for capturing the temporal dependencies of gaze directions.

**Gaze Embedding** Given the predicted gaze $\mathbf{g}_t$ at time $t$, we project $\mathbf{g}_t$ to a $d$-dimension vector $\boldsymbol{s}_t$ through a linear projection function, defined as:

$$
\boldsymbol{s}_t = \begin{cases} \boldsymbol{W}^s \cdot \boldsymbol{g}_t + \boldsymbol{b}^s, t \geq 1 \\ \boldsymbol{b}^z, t = 0 \end{cases}
\tag{6}
$$

where $t = 0$ represents the begin token and $\boldsymbol{W}^s \in \mathbb{R}^{d \times 3}$ and $\boldsymbol{b}^s \in \mathbb{R}^{d \times 1}$ represent the weight matrix and bias.

**Periodic Positional Encoding** Consider that the gaze direction could be quite consistent in a sequence, we refer to the method from Fan et al. (2022) to add a Periodic Positional Encoding (PPE) to the gaze embedding vectors, indicating the temporal order. The PPE is expressed by the function below.

$$PPE_{(t,2i)} = \sin((t \bmod P)/(10000)^{2t/d})$$
$$PPE_{(t,2i+1)} = \cos((t \bmod P)/(10000)^{2t/d}) \tag{7}$$

where the $i$ is the dimension index and $P$ is a hyper-parameter defining the period. The gaze embedding vector $\boldsymbol{s}_t$ will be added to the PPE before feeding them to the Autoregressive Transformer layer, expressed as:

$$\hat{\boldsymbol{s}}_t = \boldsymbol{s}_t + PPE(j), t = 1, \cdots, T \tag{8}$$

**Autoregressive Transformer** To model the temporal dependency of gaze movement under certain human interactions, we refer to the transformer decoder architecture used in the GPT models and design an module that autoregressively predict the gaze in one future step. Given the updated feature $\tilde{\boldsymbol{h}}_{t+1}$ from spatial model and predicted gaze direction $\boldsymbol{g}_t$ at previous time step, we model the distribution $P(\boldsymbol{g}_{t+1}|\boldsymbol{g}_t, \tilde{\boldsymbol{h}}_{t+1})$ with the devised Autoregressive Transformer $\mathcal{F}_{\Theta_3}$. In each layer of the autoregressive model, there is a Multi-Head self-attention layer (MHS) and a Multi-Head cross-attention (MHC) layer, inserted with residual connections and layer normalization(LN). The processing of our temporal model can be written as:

$$\text{self-attn: } \hat{\boldsymbol{s}}_l^{(1)} = \text{LN}(\text{MHS}(Q^{\hat{\boldsymbol{s}}_{l-1}}, K^{\hat{\boldsymbol{s}}_{l-1}}, V^{\hat{\boldsymbol{s}}_{l-1}}) + \hat{\boldsymbol{s}}_{l-1})$$

$$\text{cross-attn: } \hat{\boldsymbol{s}}_l^{(2)} = \text{LN}(\text{MHC}(Q^{\hat{\boldsymbol{s}}_l^{(')}}, K^{\tilde{\boldsymbol{h}}_l^{(1)}}, V^{\tilde{\boldsymbol{h}}_l^{(1)}}) + \hat{\boldsymbol{s}}_l^{(1)})$$
$$\text{FFN: } \hat{\boldsymbol{s}}_l = \text{FFN}(\hat{\boldsymbol{s}}_l^{(2)}), l = 1, \cdots, N_d \tag{9}$$
$$\text{Output:} \tilde{\tilde{\boldsymbol{h}}}_t = \hat{\boldsymbol{s}}_{N_d,t}$$

where $l$ is the layer index and we can concatenate $N_d$ layers in total.

**Regressing Gaze and Uncertainty** The output of the Autoregressive Transformer $\tilde{\tilde{\boldsymbol{h}}}_t$ are feed into the FC layer to regress for a probabilistic gaze prediction, described as $(\gamma, \phi, \sigma)$, where $\gamma, \phi$ is the gaze direction in sphere coordinate system and $\sigma$ represents for the gaze concentration, which reflects the gaze uncertainty. The angular formulation $\gamma, \phi$ can be converted to a unit 3D gaze vector $g = [g_x, g_y, g_z]$ by solving { $\frac{g_x}{g_z} = -tan(\gamma); g_y = sin(\phi); g_x^2 + g_y^2 + g_z^2 = 1$}. Given $\tilde{\tilde{\boldsymbol{h}}}_t$, the FC layer generate gaze prediction for $t + 1$, written as:

$$\boldsymbol{g}_{t+1}, \sigma_{t+1} = \mathcal{F}_{\Theta_4}(\tilde{\tilde{\boldsymbol{h}}}_t) \tag{10}$$

### 3.6. Human Interaction Classifier

As we are also interested in the human interaction states in the video, we build a classifier $\mathcal{F}_{\Theta_5}(\cdot)$ to generate human interaction predictions, utilizing the gaze feature $\tilde{\tilde{\boldsymbol{h}}}_t$ from Eq. 9 combined with the target for every subject. The predicted interaction category is expressed as:

$$\boldsymbol{a}_t = \mathrm{softmax}(\mathcal{F}_{\Theta_5}([\tilde{\tilde{\boldsymbol{h}}}_t; \mathrm{bbox}_t])) \tag{11}$$

When the ground-truth gaze interaction labels are given, we define a interaction loss:

$$L_{Interaction} = \frac{1}{NT} \sum_t^T \sum_i^N -\boldsymbol{a}_{t,i}^{gt} \log \boldsymbol{a}_{t,i} - (1 - \boldsymbol{a}_{t,i}^{gt}) log(1 - \boldsymbol{a}_{t,i}) \tag{12}$$

### 3.7. Loss Function

The overall training loss function for our model is :

$$Loss = \lambda_1 L_{gaze} + \lambda_2 L_{smooth} + \lambda_3 L_G + \lambda_4 L_{interaction} \tag{13}$$

where $L_G$ and $L_{Interaction}$ are gaze interaction graph loss and interaction classification loss, as defined in Eq. 4 and Eq. 12. We define $L_{gaze}$ and $L_{smooth}$ below.

We compute the negative log-likehood loss with the predicted gaze angle $\boldsymbol{g}_t = [\boldsymbol{\gamma}, \boldsymbol{\phi}]$ and uncertainty $\boldsymbol{\sigma}_t$, when the ground-truth gaze labels are given. The gaze loss $L_{gaze}$ is defined as:

$$L_{gaze} = \frac{1}{T} \sum_t^T (\log(\boldsymbol{\sigma_t}) + \frac{1}{\boldsymbol{\sigma}_t} \|\boldsymbol{g}_t - \boldsymbol{g}_t^{gt}\|_2) \tag{14}$$

We also impose a smoothness constraint along the temporal axis, to minimize the difference between the predicted gaze in two consecutive frames, which is formulated as:

$$L_{smooth} = \frac{1}{T-1} \sum_{t=1}^{T-1} \|\boldsymbol{g}_t - \boldsymbol{g}_{t+1}\|_2 \tag{15}$$

### 3.8. Training Algorithm

As gaze benchmark dataset and human gaze interaction dataset do not have intersecting labels, we propose a multi-state algorithm to train the full model.

- Stage 1: temporal model pre-training (the blue part in Fig. 3). We use the gaze benchmark datset that contains image sequences/videos and frame-by-frame gaze annotations to pre-train the feature extractor and the temporal model $(\mathcal{F}_{\Theta_1}(\cdot), \mathcal{F}_{\Theta_3}(\cdot), \mathcal{F}_{\Theta_4}(\cdot))$;

- Stage 2: Freeze $\mathcal{F}_{\Theta_3}(\cdot)$ and train spatial model (the orange part in Fig. 3). We use human interaction dataset that contains image sequences/videos and frame-by-frame interaction annotation to train the feature extractor, structure-aware Transformer and the interaction classifier, i.e., $(\mathcal{F}_{\Theta_1}(\cdot), \mathcal{F}_{\Theta_2}(\cdot), \mathcal{F}_{\Theta_4}(\cdot), \mathcal{F}_{\Theta_5}(\cdot))$;

- Stage 3: Full model training on mixture data.

We show the training algorithm in Algorithm. 1.

---

**Algorithm 1** Multi-Stage Training Process

---

1: **Stage1: pre-training temporal model with Gaze annotation**
2: Input: $\boldsymbol{I}_{1:T}$ ; Labels: $\boldsymbol{g}_{1:T}^{gt}$
3: Parameters to learn: $\mathcal{F}_{\Theta_1}(\cdot), \mathcal{F}_{\Theta_3}(\cdot), \mathcal{F}_{\Theta_4}(\cdot)$
4: Training Loss: $L_1 = \lambda_1 L_{gaze} + \lambda_2 L_{smooth}$
5: **Stage2: interaction-aware training with gaze to interaction graph**
6: Input: $\boldsymbol{I}_{1:T}$ ; Labels: {gaze interaction category $\boldsymbol{a}_{1:T}^{gt}$, bbox$_{1:T}$, Graph edges $\boldsymbol{e}_{ij}^{gt}$ }
7: Parameters to learn: $\mathcal{F}_{\Theta_1}(\cdot), \mathcal{F}_{\Theta_2}(\cdot), \mathcal{F}_{\Theta_4}(\cdot), \mathcal{F}_{\Theta_5}(\cdot)$
8: Training Loss: $L_2 = \lambda_2 L_{smooth} + \lambda_3 L_G + \lambda_4 L_{interaction}$
9: **Stage3: full model training**
10: Input: mixture of gaze benchmark sequences and gaze interaction sequences (semi-supervised)
11: Parameters to learn: $\mathcal{F}_{\Theta_1}(\cdot), \mathcal{F}_{\Theta_2}(\cdot), \mathcal{F}_{\Theta_3}(\cdot), \mathcal{F}_{\Theta_4}(\cdot), \mathcal{F}_{\Theta_5}(\cdot)$
12: Training Loss: $L_3 = \lambda_1 L_{gaze} + \lambda_2 L_{smooth} + \lambda_3 L_G + \lambda_4 L_{interaction}$

---

| Methods | Gaze360 (frontal) | Gaze360 (full) |
|---|---|---|
| RT-Gene Fischer et al. (2018) | 12.26 | - |
| Dilated-Net Chen and Shi (2018) | 13.73 | - |
| CA-Net Cheng et al. (2020) | 12.26 | - |
| Gaze360 Kellnhofer et al. (2019) | 11.1 | 13.5 |
| LAEO Kothari et al. (2021) | 10.1 | 13.2 |
| L2CS Abdelrahman et al. (2022) | 10.41 | - |
| **Dyn-Gaze(ours)** | **10.03** | ***11.27** |

Table 1: Within-dataset evaluation on Gaze360 dataset.

## 4. Experiments

**Datasets.** We investigate model performance on three sequence-based benchmark datasets: Gaze360Kellnhofer et al. (2019) , EyeDiap Funes Mora et al. (2014) and VACATION Fan et al. (2019). 1) Gaze360 contains in-the-wild human images captured by a 360° camera with a wide range of horizontal gaze direction. Large amount of images in Gaze360 is quite blurred or self-occluded due to large head pose, hence making it a challenging dataset for detecting eye gaze direction. Only one subject appears in the scene for every sequence. 2) EyeDiap is a video-based dataset recording a participant head and eye movement when tracking a static or a moving target. Only one subject appears in the scene for every video. 3) VACATION is a video dataset that covers diverse social scenes and complete gaze communication annotations. For every video, there can be multiple people communicating with each other or looking at different objects. The bounding box for the subjects/objects appeared in the scene are given.

**Implementation Details** We set the sequence length $T = 7$ and sample short sequences by sliding window on a video. The number of transformer layers $N_d = 2$. The hyperparameter $\alpha = 4$ in Eq. (2), the training weights are set as $\lambda_1 = 5, \lambda_2 = 0.10, \lambda_3 = 1, \lambda_4 = 2$. We train 80 epoches in Stage 1, 50 epoches in Stage 2 and 50 epoches in Stage 3.

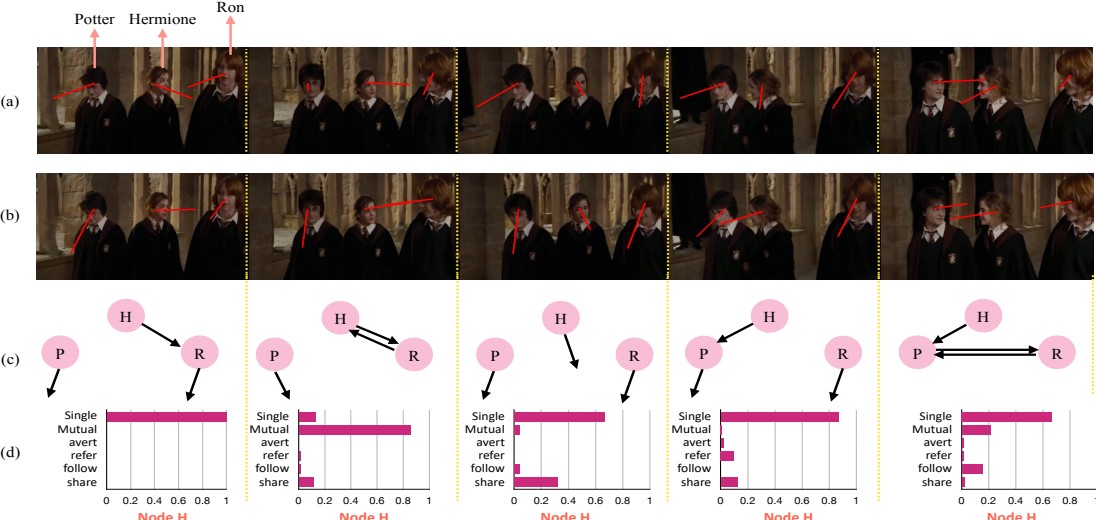

Figure 6: Visualization of predicted gaze direction and gaze interaction probability. Three people "Potter", "Hermione", "Ron" are interacting in the scene. **Row-a**: the predicted gaze from the pre-trained gaze estimator (without involving interaction information in training). **Row-b**: predicted gaze from fine-tuned gaze estimator with interaction labels. **Row-c**: the Human Interaction Graph constructed from the predicted gaze. **Row-d**: the predicted probability of interaction class for Node H.

### 4.1. Within-Dataset Evaluation

**Gaze estimation** To prove that our proposed geometric constraints improve gaze estimation performance, we first perform within-dataset evaluation on Gaze360 and Eye-Diap, as shown in Table 1. We compare our method, the Gaze-Geo and UGaze-Geo, with SOTA learning-based methods, including RT-Gene (Fischer et al. (2018)), Dilated-Net (Chen and Shi (2018)), CA-Net (Cheng et al. (2020)), Gaze360 (Kellnhofer et al. (2019)), LAEO (Kothari et al. (2021)) and L2CS Abdelrahman et al. (2022). On Gaze360 we use the official train-val-test set division and present the evaluation results on different ranges of gaze directions, including frontal faces (column 2 in Table 1) and all faces (column 3 in Table 1). As we know, LAEO Kothari et al. (2021) only consider the "looking at each other" constraints, which is one type of the gaze interactions. Our model considers multiple types of gaze communication activities and the dynamic state transitions among them. Our method outperforms other methods on Gaze360, especially on the full range of evaluation set Gaze360(full), our model reduces the gaze angular error by 14.6% comparing with LAEO (Kothari et al.). Our dynamic gaze estimation model also improves the performance on frontal poses slightly compared to Kothari et al. (2021).

**Gaze Interaction Classification** We also perform an evaluation of the gaze interaction classification accuracy, compared with Fan et al. (2019). For the six types of gaze communications, we calculate the precision and F1 score. As shown in Fig. 2, our model has

| Task | single | mutual | avert | refer | follow | share | Avg. |
|---|---|---|---|---|---|---|---|
| Fan et al. (2019) | 26.17 | 98.60 | 74.28 | 53.16 | 18.05 | 46.61 | 55.02 |
| Dyn-Gaze (Ours) | 38.12 | 90.27 | 68.79 | 55.67 | 40.12 | 54.50 | 57.92 |

Table 2: F1-score of gaze communication prediction on VACATION dataset.

bettwe performance on the task of classifying gaze interactions between two subjects. In terms of F1 score, our dynamic gaze model improves the prediction accuracy by 2.8% on average compared to Fan et al. (2019). In addition, we can also observe the advantages of utilizing gaze direction directly for analyzing specific gaze interactions, such as "single" (($\uparrow$ 11.95%) ), "follow" ($\uparrow$ 22.07%) , and "share" ($\uparrow$ 7.89%) . In Fig. 6, we show an example of gaze & interaction prediction on a test video of VACATION dataset. We compare the gaze direction before & after applying the interaction-based fine-tuning and we can see evident improvement in the qualitative results. We also show the constructed Human Interaction Graph using the gaze direction, as described in Eq. 3.

### 4.2. Cross-Dataset Evaluation

We also conduct a cross-dataset experiment to elaborate that, by modeling the spatial gaze relationships and temporal dependency, our gaze estimator is robust and have better generalization ability under large data difference. Following the cross-data settings adopted in existing works PureGaze Cheng et al. (2022) and RAT Bao et al. (2022), we train our model on Gaze360 and ETH-XGaze and then evaluate on EyeDiap. In Table 3 we compare the performance of our model with SOTA gaze estimation methods, including RT-Gene, Dilated-Net, CA-Net, FullFace, Gaze360, PureGaze and RAT.

As shown in Table 3, our model can achieve SOTA cross-dataset performance when testing on new video dataset. Compared to the dynamic model Gaze360 (Kellnhofer et al. (2019)) that only predicts the gaze for the central frame, our model can significantly reduce the gaze error on EyeDiap dataset by 57.8%. Compared to the 2D data augmentation method Bao et al. (2022), our gaze model benefit from the spatial constraints constructed based on the subject interactions, especially when one subject is fully occluded due to large head pose. Our model outperforms Bao et al. (2022) by reducing the cross-dataset gaze error by 5.9%.

### 4.3. Ablation Study

We perform an ablation study to validate the effectiveness of our training algorithm for each stage for gaze estimation. In Table. 4, we analyze the within- and cross-dataset performances with different combinations of spatial and temporal modules. We explore five different combinations listed as below.

- Static Model: ResNet feature extractor ($\mathcal{F}_{\Theta_1}$) + FC layer($\mathcal{F}_{\Theta_4}$), trained on gaze images.

- Static + Interaction: ResNet feature extractor ($\mathcal{F}_{\Theta_1}$) + Structure-aware Transformer($\mathcal{F}_{\Theta_2}$) + Interaction classifier($\mathcal{F}_{\Theta_5}$) + FC layer($\mathcal{F}_{\Theta_4}$), trained on gaze images and gaze interaction images.

| Methods | Gaze360 $\rightarrow$ EyeDiap |
|---|---|
| FullFace Zhang et al. (2017) | 14.42 |
| RT-Gene Fischer et al. (2018) | 38.60 |
| Dilated-Net Chen and Shi (2018) | 23.88 |
| Gaze360 Kellnhofer et al. (2019) | 11.86 |
| CA-Net Cheng et al. (2020) | 31.41 |
| PureGaze Cheng et al. (2022) | 9.32 |
| Res-Net18+RAT Bao et al. (2022) | 7.10 |
| **Dyn-Gaze (ours)** | **6.68** |

Table 3: Cross-dataset evaluation from Gaze360 to EyeDiap and comparision with SOTA learning-based methods

| Models | Within-data | | | Cross-data |
|---|---|---|---|---|
| | Gaze360 (frontal) | Gaze360 (full) | EyeDiap | Gaze360 $\rightarrow$ EyeDiap |
| Static Model | 10.54 | 13.55 | 4.89 | 7.23 |
| Static + Interaction | 10.25 | 12.18 | 4.86 | 7.31 |
| Interaction only | 28.8 | 30.12 | 15.15 | - |
| Temporal | 10.24 | 12.98 | 4.44 | **6.60** |
| Temporal + Interaction (Full) | **10.03** | **11.27** | **4.25** | 6.68 |

Table 4: Ablation study of gaze angular errors when applying different constraints and w/o uncertainty modeling during training. The last two rows are corresponding to Gaze-Geo and UGaze-Geo.

- Interaction only: ResNet feature extractor $(\mathcal{F}_{\Theta_1})$ + Structure-aware Transformer$(\mathcal{F}_{\Theta_2})$ + Interaction classifier$(\mathcal{F}_{\Theta_5})$ + FC layer$(\mathcal{F}_{\Theta_4})$, trained on gaze interaction images (weakly supervised training).

- Temporal: ResNet feature extractor $(\mathcal{F}_{\Theta_1})$ + Autoregressive Transformer$(\mathcal{F}_{\Theta_3})$ + FC layer$(\mathcal{F}_{\Theta_4})$, trained on gaze sequences.

- Temporal + Interaction: our final model, trained on gaze sequences and gaze interaction sequences.

The Static model is the baseline model trained with full-supervision on gaze benchmark dataset, without considering the spatial interaction or temporal dependency. By comparing Static Model with Static + Interaction model, we show that utilizing multi-subject gaze interaction can help to refine the gaze estimator, especially for the full pose cases. However, only using gaze interaction labels (without any gaze label supervision) will fail to generate reliable gaze estimation results. By comparing the Static Model with the Temporal Model, we show the effectiveness to consider the temporal dependency of gaze movement, as we can observe significant performance improvement on both within- and cross-dataset experiments. Our final model that learns interaction dependency and temporal dependency achieves the best within dataset performances on both Gaze360 and EyeDiap with significant improvement compared to the Static Model.

## 4.4. Conclusion

In this paper we propose a framework to perform interaction-aware dynamic gaze estimation in videos, which utilize the gaze communication labels among multiple subjects/objects and the temporal dependency of gaze movement to refine the gaze estimator. Specifically, we define a direct mapping from predicted 3D gaze direction to human gaze interaction types and construct a Human Interaction Graph based on gaze and bounding box locations. We perform the dynamic gaze prediction in an auto-regressive manner, by modeling the future gaze distribution conditioned on current gaze and human interaction graph structure. Our model fully utilize the dataset without gaze annotations and propose a multi-stage training algorithm to alternately updating the temporal gaze prediction module and gaze interaction module. In terms of performances, we proved that by introducing the interaction constraints and temporal constraints, our model can be significantly improved compared to the static model on video dataset.

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
