# OpenReview forum: "Interaction-aware Dynamic 3D Gaze Estimation in Videos"
_NeurIPS.cc/2023/Workshop/Gaze_Meets_ML — Gaze Meets ML 2023 Oral_

### Official Review · Reviewer_naLH · 2023-10-18
**The paper focuses in the domain of dynamic gaze estimation, particularly emphasizing the utilization of human interaction labels to enhance the model's performance.**

**Rating:** 7
**Confidence:** 4

**Review:**

The author address the challenging task of dynamic 3D gaze estimation in real-world video scenarios, particularly focusing on the dynamics of human gaze interactions. The abstract effectively conveys the paper's contributions in the field of gaze dynamics, highlighting the complexities associated with annotating gaze in videos and the limitations of existing methods. The proposed approach leverages human interaction labels to enhance gaze estimation, which is a notable and innovative contribution.

The introduction provides a comprehensive background, discussing the importance of eye gaze in various applications and the evolution of deep learning models in gaze estimation. It also clearly discusses prior efforts to capture gaze dynamics in videos, emphasizing the difficulty of obtaining accurate annotations for training. The review of related work on using secondary labels such as point of gaze and human interaction information to refine gaze estimation is informative and helps contextualize the research.

The paper's unique contribution is clearly stated: a dynamic 3D gaze estimation framework that considers various human gaze interaction activities and their transitions. The use of a Human Interaction Graph and a Transformer-based spatiotemporal module to capture spatial and temporal relationships in gaze movements is an interesting approach. Furthermore, the multi-stage training algorithm is designed to jointly update the interaction and gaze estimation modules, improving the overall gaze estimator

Overall, the work demonstrated improvements in both within-domain and cross-domain gaze estimation accuracy on the Gaze360 benchmark and highlighted the practical significance of the proposed method. However, In Eq. 1,  P(a_t |g_t) was modeled with learning a Human Interaction Graph G_t from g_t; then  P(a_(t+1)|g_t, a_t, I_(t+1)) was modeled with a Structure-aware, these terms P(a_t |g_t) and P(a_(t+1)|g_t, a_t, I_(t+1)) were not clearly stated what theystand for. Also, what the interaction score C_ij signify when it is greater than one was not discussed.

---

### Official Review · Reviewer_dhU2 · 2023-10-20
**Interesting method using Autoregressive Transformer for Gaze estimation in the wild**

**Rating:** 9
**Confidence:** 4

**Review:**

The manuscript presents a strategy to use Autoregressive Transformers to estimate gaze in video sequences captured of subjects in the wild. The proposed framework performs interaction-aware dynamic gaze estimation in videos that combine gaze communication labels with the temporal dependency of gaze movement to refine the gaze estimator. They construct a Human Interaction Graph based on gaze and bounding box locations to perform dynamic gaze prediction in an auto-regressive manner, by modeling the future gaze distribution conditioned on current gaze and human interaction graph structure.

They also present a thorough literature review and use various data sets to compare the performance of their method against existing state of the art results. Their methods demonstrate performance improvements through the statistical significance of the gains has not been measured. That said, the goal of their work is to reduce the burden in labeling and any improvement is welcome. Further, they also test on datasets where they utilize a dataset without gaze annotations and propose a multi-stage training algorithm to alternatively updating the temporal gaze prediction module and gaze interaction module.

The work is clearly described. However, it is unclear if their code is available for reproduction and objective evaluation by others.

---

### Official Review · Reviewer_bfCQ · 2023-10-23
**The authors of the paper "Interaction-aware Dynamic 3D Gaze Estimation in Videos" address an important aspect, namely the**

**Rating:** 9
**Confidence:** 3

**Review:**

Despite the multitude of studies in gaze estimation, gaze dynamic estimation/learning has not address yet in full length in the literature. Capturing gaze in videos where the subject has frequent head and body movements poses a challenge. Similarly, learning some type of gazing might not be applicable (see generalization)  to other type of data as the model might be biased toward a certain type of data.

In order to address these issues the authors are proposing a a dynamic 3D gaze estimation framework, a spatio-temporal model to properly model the interaction between subjects and their gaze and finally, a Human Interaction model/graph to further enhance the gaze estimator.

The paper is written nicely, everything is explained, the different figures (see Fig 2, 3, 4) explain the problem and the overall model. The Related Work part is concise but capture the  main idea around the dynamic gaze estimation. The problem at hand is formulated properly, and the model is described in details.

The benchmark datasets (see 3) are described and comparative results are provided, showing a certain type of superiority, though no significance test was applied. The comparative results mentioned in Table 3 also show some superiority of the current solution. One could say that the reviewers were paying attention to the evaluation, and the results are rather promising.

The only criticism would be that the ResNet, AutoRegressive Transformers were not properly described.

---

### Meta-Review · Area_Chair_SVtS · 2023-10-26

**Recommendation:** Accept (Oral)
**Confidence:** 4

**Metareview:**

This paper introduces a novel approach to dynamic 3D gaze estimation in videos, leveraging human interaction labels and Autoregressive Transformer structures. The method unifies a human-interaction graph with a Transformer-based spatial and temporal module to comprehensively model spatial and temporal relationships in human gaze movements.
Reviewers have lauded the paper for its clear presentation and notable contributions. Nonetheless, reviewers raised some concerns, such as regarding the availability of the code, which the authors could address in the final version. Overall, this is a well-written paper, recommending acceptance.

---

### Decision · Program_Chairs · 2023-10-26

Accept (Oral)